# Persistent Topological Features in Large Language Models

Yuri Gardinazzi*[1][2]  Karthik Viswanathan*[1][3]  Giada Panerai[1][2]
Alessio Ansuini[1]  Alberto Cazzaniga[1]  Matteo Biagetti[1]

## Abstract

Understanding the decision-making processes of large language models is critical given their widespread applications. To achieve this, we aim to connect a formal mathematical framework—zigzag persistence from topological data analysis —with practical and easily applicable algorithms. Zigzag persistence is particularly effective for characterizing data as it dynamically transforms across model layers. Within this framework, we introduce topological descriptors that measure how topological features, $p$-dimensional holes, persist and evolve throughout the layers. Unlike methods that assess each layer individually and then aggregate the results, our approach directly tracks the full evolutionary path of these features. This offers a statistical perspective on how prompts are rearranged and their relative positions changed in the representation space, providing insights into the system's operation as an integrated whole. To demonstrate the expressivity and applicability of our framework, we highlight how sensitive these descriptors are to different models and a variety of datasets. As a showcase application to a downstream task, we use zigzag persistence to establish a criterion for layer pruning, achieving results comparable to state-of-the-art methods while preserving the system-level perspective.

## 1. Introduction

Large Language Models (LLMs) have revolutionized natural language processing by achieving unprecedented performance levels across a wide range of tasks (see Raiaan et al. (2024) for a review). Despite their success, the black-box nature of these models has raised significant concerns

about interpretability and transparency (Liao & Vaughan, 2023). Moreover, their large scale demands a considerable amount of computational resources (Samsi et al., 2023; Bai et al., 2024), making it essential to reduce their size without compromising performance (Ma et al., 2023; Gromov et al., 2024; Men et al., 2024).

One strategy for addressing these issues has been to study the models' internal representations. Early works (Zeiler & Fergus, 2014) demonstrated that visualization techniques can effectively uncover hierarchical representations within convolutional neural networks, highlighting how lower layers focus on edge detection while higher layers correspond to object parts and semantic concepts. Additionally, (Olah et al., 2018) illustrated that analyzing weight matrices and neuron activations can reveal interpretable features and organizational structures within deep networks, providing insights into how complex patterns are encoded and processed.

More recently, geometric studies made progress by introducing concepts like intrinsic dimension to characterize the manifold of internal representations and its evolution across layers (Ansuini et al., 2019; Doimo et al., 2020; Pope et al., 2021). These methods have been successfully applied to transformer models in various works (Valeriani et al., 2023; Tulchinskii et al., 2024; Cheng et al., 2023; 2024; Viswanathan et al., 2025). One notable achievement of this approach has been to show the emergence of semantic knowledge and abstraction phases in the middle layers of models, rather than at the final layers, as might be intuitively expected.

However, these approaches provide only a static view of internal representations and suffer limitations in tracking their changes across layers.

A natural framework to address these limitations and to offer a more comprehensive characterization of the geometry of internal representations of neural networks is Topological Data Analysis (TDA). TDA is a set of unsupervised techniques that offers robust methods to describe the shape and structure of complex datasets. It has seen exponential growth with applications in computational biology (Mandal et al., 2020), cosmology (Biagetti et al., 2021; Yip et al., 2024)], personalized medicine (Skaf & Laubenbacher, 2022), time-dependent data analysis (El-Yaagoubi et al.,

---

*Equal contribution  [1]Area Science Park  [2]University of Trieste  [3]University of Amsterdam. Correspondence to: Matteo Biagetti <matteo.biagetti@areasciencepark.it>.

*Proceedings of the 42nd International Conference on Machine Learning*, Vancouver, Canada. PMLR 267, 2025. Copyright 2025 by the author(s).

2023), and machine learning (Hensel et al., 2021), just to name a few. One prominent tool within TDA is persistent homology, which tracks the birth and death of topological features across different scales, thereby capturing the multiscale behavior of a point cloud. Several studies have proposed persistent homology to investigate neural networks and their internal representations (e.g. (Rieck, Bastian Alexander et al., 2023), (Naitzat et al., 2020; Lacombe et al., 2021; Magai & Ayzenberg, 2022)).

However, in the context of TDA applications, it has not yet been recognized that the internal representations of neural networks can essentially be viewed as point clouds dynamically evolving in time (layers). In the particular case of LLMs, as pre-trained models process inputs, they transform these point clouds within the representation space layer by layer, capturing essential features and relationships throughout the model's depth. Thus, it is natural to interpret these transformations as an evolving discrete dynamical system. To address this problem, we exploit a TDA tool developed to characterize time-varying point clouds and temporal networks, known as *zigzag persistence*.

Our approach achieves the following results:

- **ZigZag Framework for LLMs:** We build a fast and scalable pipeline to characterize the birth and death of topological features across transformer models' layers. As new contributions in the context of zigzag applications, we introduce the k-Nearest Neighbors-based filtration, and we interpret layers as time snapshots, tracking the trajectory of features across layers.

- **Identification of Phases of Prompt Processing:** Using interpretable topological descriptors, we characterize the model's dynamical processing of prompts in representation space across layers. We use this characterization to identify four distinct phases as prompts move in representation space across layers: an initial phase with rapid rearrangement of positions, a middle phase characterized by stable, long-lived relations among prompts, a transition phase where the model refines these relations, and a final phase of new rearrangements preparing data for output.

- **Model Pruning:** As a showcase downstream task, we use our descriptors to define a criterion to prune layers without significantly degrading performance, finding comparable results to state-of-the-art methods.

Our topological descriptors show quantitatively different results but qualitatively similar across models, datasets, and choices of zigzag hyperparameters. This proves their expressivity and simultaneously shows a degree of universality in the topological structure of LLM representations.

## 2. Related Work

**Topology of Internal Representations.** TDA has been extensively used in machine learning (see (Hensel et al., 2021) for a recent review). In the context of studying internal representation, studies on Convolutional Neural Networks (CNN) used topological descriptors to explore the shape of activation functions (Rathore et al.) or their relations to performance (Naitzat et al., 2020). (Magai & Ayzenberg, 2022) introduced persistent homology dimension as an estimator of the intrinsic dimension of internal representations in CNNs, while (Barannikov et al., 2022) proposed a measure of similarity based on topological descriptors to compare representations. Betti numbers have been observed to remain stable across different datasets for the same architectures and to decrease as depth increases (Suresh et al., 2023).

**Zigzag Persistence.** Zigzag persistence was introduced in (Carlsson & de Silva, 2010; Carlsson et al., 2009; Tausz & Carlsson, 2011) as an extension of persistent homology to study the persistence of topological features across sequences of spaces. This approach is particularly useful when data undergo dynamic changes or transformations over time. Since its introduction, zigzag persistence has been applied in various fields, including Hopf bifurcations in dynamical systems (Tymochko et al., 2020), commuting patterns in Great Britain's transportation network Myers et al. (2023), coral reef ecosystems (McDonald et al., 2023), cell location time series (Yang et al., 2023; Zhang et al., 2023), and honeybee aggregations (Gharooni-Fard et al., 2024). It has also inspired methodological extensions such as multidimensional persistence (Kim & Mémoli, 2021) and the development of formigrams and crocker stacks (Xian et al., 2022).

**Distinguishing Transformer Stages.** Recent works have studied the geometry of internal representations to identify distinct stages[1] in the way pre-trained transformer models process input across layers. For instance, Valeriani et al. (2023) demonstrated that transformer models exhibit a phase transition in the middle layers, characterized by a spike in intrinsic dimensionality, which correlates with the emergence of semantic and syntactic abstractions. Similarly, Cheng et al. (2023) highlighted that middle layers play a crucial role in compressing input representations into lower-dimensional manifolds, enabling the model to generalize and handle complex linguistic tasks. Subsequent work has confirmed and expanded these findings in different ways (Lad et al., 2024; Artzy & Schwartz, 2024; Skean et al., 2024).

**Layer Pruning in Large Language models.** Among ex-

---

[1]While previous work has frequently used the word "stage" in this context, we prefer "phase" to emphasize the continuous, evolving nature of the process.

isting methods to reduce the size of neural networks, layer pruning has gained particular relevance in the context of LLMs. The first applications to BERT models (Fan et al., 2020; Zhang & He, 2020; Fan et al., 2021; Jha et al., 2024) inspired a long series of experiments employing similar techniques (Sajjad et al., 2023; Siddiqui et al., 2024; He et al., 2024; Zhang et al., 2024a; Kim et al., 2024; Zhang et al., 2024b). Many of these efforts base their methodology on similarity measures of internal representations, which have conveniently been summarized in a recent review (Klabunde et al., 2023). In this work, we consider (Gromov et al., 2024), which uses angular similarity, and (Men et al., 2024), which uses Block-Influence similarity, as a reference point for comparison.

## 3. Method

In this section, we introduce the zigzag persistence framework, which we use to analyze the internal representations of LLMs pre-trained with an autoregressive loss. These models typically receive an input sequence of $n$ tokens (often representing a sentence) embedded in a $d$-dimensional space. The input is transformed across the network layers without altering the embedding dimension. Due to the autoregressive nature of these models, the representation of the last token in a sequence captures information about the entire sequence and is used to predict the next. As a result, we choose to focus on the last token representation of each sequence at each layer. Thus, our point cloud is represented by last tokens embeddings, i.e. vectors of the form $\{\mathbf{x}_i(\ell_j)\} \in \mathbb{R}^d$, for $i = 1, ..., N_{\text{sentences}}$ and $j = 1, ..., N_{\text{layers}}$. These last tokens are extracted from a variety of datasets and serve as an observational probe of how the model processes input.

### 3.1. Topological Data Analysis

Topological data analysis (Edelsbrunner et al., 2002; Zomorodian & Carlsson, 2004) provides a tool for geometrically characterizing highly complex datasets. Within this framework, persistent homology (Carlsson, 2009) is the key methodology to characterize a point cloud on multiple scales at once. Its goal is to identify the range of scales over which a particular class of topological features (connected components, loops, voids, higher dimensional "holes") remain relevant, or "persistent", as opposed to "topological noise", i.e. features disappearing roughly at the same scale they formed. The basic ingredients for this technique are i) a criterion to connect points, forming a *simplicial complex* and ii) a scale parameter $\nu$ (often a coarsening scale) such that given $\nu_1 \leq \nu_2$, then the two corresponding simplicial complexes are related by $K_{\nu_1} \subseteq K_{\nu_2}$. The ordered sequence of simplicial complexes for varying scale parameters is called *filtration*. An intuitive example is the Vietoris-Rips filtration, built from complexes parametrized by the radius of the

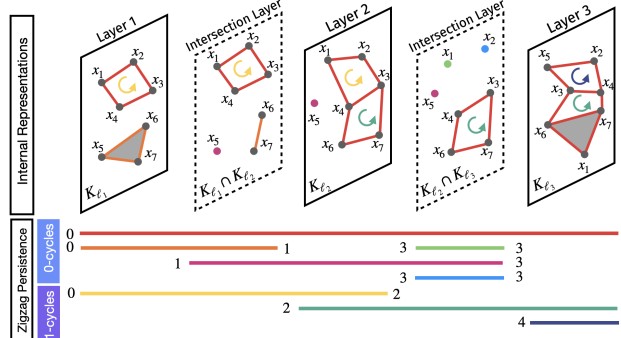

*Figure 1.* A schematic representation of the zigzag algorithm. It shows how zigzag can track the evolution of topological features over time, and the descriptors of this work are built upon them.

ball drawn around each point of the dataset.

Filtrations can be generalized to a more flexible structure called a *zigzag filtration*. Unlike a standard filtration, a zigzag filtration allows the sequence of complexes to move both forward and backward, meaning that inclusions between complexes can reverse at certain steps. We take this approach in our study to track the evolution of the internal representations *across* layers, rather than at a fixed snapshot, as done in traditional persistent homology implementations. In this sense, our parameter is not a distance/coarsening scale, but a discrete *time* scale represented by the layer number. We track topological features as they are formed and destroyed along the model layers and statistically characterize these changes to describe a complex series of transformations in high-dimensional space. Differently than standard persistent homology, short- and long-lived features represent how the model dynamically evolves. Short-lived features indicate a high rate of rearrangement of the points between adjacent layers, while long-lived features suggest a phase of retention of (relative) positions across several layers. This is a crucial point in our analysis, as it provides a novel tool to interpret how the model processes different inputs by moving them and changing their relative positions in the representation space. Recent computational advances have allowed feasible implementation of these methods for analyzing time-varying point clouds within the broader machine learning community. For instance, efficient algorithms such as the fast zigzag persistence method introduced by Dey & Hou (2022) have enabled scalable analysis of evolving topological features, making persistence-based approaches much more accessible in large-scale applications. We now outline the main steps of the zigzag algorithm, leaving a rigorous mathematical formulation to Appendix A.

## 3.2. Zigzag Persistence for Layer Analysis

We aim to study internal representations by tracking statistical changes in the formation of $p$-dimensional holes generated by connecting nearby data points within each layer $\ell_i$. As introduced above, the first ingredient for a TDA formulation is a criterion for connecting points. In this regard, we construct a k-Nearest Neighbors graph $G_{\ell_i} = (V_{\ell_i}, E_{\ell_i})$ at every layer $\ell_i$, where the number $k_{NN}$ of neighbors is a fixed hyperparameter (see (Le & Taylor, 2024) for a previous use of a $k_{NN}$-based filtration). To explore higher-order relations among points, we extend the dimension of the graph by filling higher-dimensional simplices. More precisely, we *fill* a simplex when its boundary, composed of lower-dimensional simplices (such as vertices and edges), is complete. In particular, we consider a triangle as filled when it has three vertices with pairwise connections. Similarly, a tetrahedron is filled when four vertices are all interconnected by edges, totaling six edges. This concept extends to higher dimensions up to a specified maximum dimension $m$. Thus, in each layer, we construct the simplicial complex $K_{\ell_i}$ defined by:

$$K_{\ell_i} = \bigcup_{S \subseteq V_{\ell_i}} \big\{ S \mid \forall x_s, x_l \in S, \ (x_s, x_l) \in E_{\ell_i}$$
$$\text{and } |S| \leq m + 1 \big\}. \quad (1)$$

To track changes in the network, we compute intersection layers by identifying simplices present simultaneously in both adjacent layers. This allows us to construct a sequence of inclusions between these complexes

$$K_{\ell_1} \leftarrow \quad \nearrow K_{\ell_2} \leftarrow \quad \cdots \quad \nearrow K_{\ell_{L-1}} \leftarrow \quad \nearrow K_{\ell_L} \quad (2)$$
$$K_{\ell_1} \cap K_{\ell_2} \qquad \cdots \qquad K_{\ell_{L-1}} \cap K_{\ell_L}$$

where we define $L \equiv N_{\text{layers}}$ for conciseness. This sequence represents our zigzag filtration, denoted by $\Phi$. This filtration is the second ingredient needed to define persistent homology. We thus define a notion of *birth* and *death* of $p$-dimensional holes, with $p = 0, ..., m - 1$, being $m$ the maximum dimension at which we expand the graph. Throughout this work, we choose $m = 4$, which implies that the $p$-dimensional holes are well defined up to dimension $p = 3$. We can track the persistence of these objects as they appear in a given layer when a group of points exhibits a particular proximity and distribution in the complex and disappear at a subsequent layer when some points have moved apart, causing them to vanish. We illustrate the idea in Figure 1.

**Comparison to similar frameworks.** A distinguishing feature of our methodology is the choice of a $k_{NN}$ filtration, whose stability was discussed in (Le & Taylor, 2024) in the context of persistent homology, though it was never applied

to zigzag persistence. A notable effort in describing spatio-temporal networks similarly to this work is (Kim & Mémoli, 2021), where the main summary statistic (the rank invariant) involves calculating a 6-dimensional data vector (4 across layers and 2 across scale) and thus combines a variation of both a time and a scale parameter, using the Rips filtration. Varying also the scale parameter is worth investigating in this context, and the techniques in (Kim & Mémoli, 2021) would be a starting point for implementing it. The work (Kim & Mémoli, 2023) from the same authors is also related to our work since the maximal group diagram and the persistence clustergram (cfr. Figure 2) are "annotated" (with the representative topological features) barcodes. In this work, they fix a scale similar to our case.

**Zigzag Persistence Diagram.** The output of the zigzag algorithm is then a multiset of birth-death pairs $[b, d]$[2], known as the *persistence diagram*

$$\text{Pers}_p(\Phi) = \Big\{ [b, d] \mid b, d \in \{0, \ldots, 2(N_{\text{layers}} - 1)\} \Big\}. \quad (3)$$

We thus work with a zigzag filtration naturally indexed by $\{0, 1, 2, \ldots, 2(N_{\text{layers}} - 1)\}$. Specifically, as shown in the Figure 1, even numbers starting from 0 are assigned to $p$-dimensional holes that emerge and disappear within the model layers. In contrast, odd numbers are designated for features at the intersection layers. It is important to note that homology classes are defined as equivalence classes, meaning that a connected component (in the case of 0-dimensional homology) need not maintain the same form at the level of simplices throughout its lifetime. The orange connected component in the figure exemplifies this: in Layer 1, it corresponds to the three points $\{x_5, x_6, x_7\}$ connected by edges, forming a triangle. In the intersection layer, it is reduced to the edge $\{x_6, x_7\}$. In Layer 2 this edge merges with another connected component (depicted in red), marking the death of the orange component. This feature ensures the robustness of our construction to small changes in the $k_{NN}$ graph. A mathematical explanation of this is provided in Appendix A. The algorithm that generates $\text{Pers}_p(\Phi)$ is schematically described in Appendix B, and in Appendix C we show a toy example using a calendar month task to visualize how we track zigzag barcodes.

**Effective Persistence Image.** The pairs generated within $\text{Pers}_p(\Phi)$ can be visualized through a *persistence image*, a well-known descriptor within the TDA tools. The persistence image in our case results in a grid of size $(2N_{\text{layers}} - 1) \times (2N_{\text{layers}} - 1)$, for each homology dimension $p$. Each pixel in the grid is associated with an integer

---

[2]The repetition of a pair $[b, d]$ indicates that multiple holes in dimension $p$ have been created and destroyed in correspondence of the same layers.

value corresponding to the number of holes appearing with that birth-death pair. Defined this way, the persistence image does not discriminate between the model and intersection layers. Their behavior is generally fairly different, and have an alternating structure between model and intersection layers. Hence, persistence images are not *smooth* as a function of layers. To achieve a smoother representation, we introduce *effective persistence images*, obtained by excluding the intersection layers from the construction. This is achieved by defining a map, similar to the approach in (Kim & Mémoli, 2017), that translates the collection of intervals from the zigzag persistence diagram of the filtration in (2) into intervals, where the birth and death occur only across model layers. Formally, for $b, d > 0$, we obtain:

$$\widehat{PI}_p(b/2, d/2) = PI_p(b, d) + PI_p(b - 1, d) \\ + PI_p(b, d - 1) + PI_p(b - 1, d - 1),$$
(4)

where $\widehat{PI}_p$ is the effective persistence image for the $p$-dimensional holes and $b, d$ are model layers indexed by even numbers.[3]

### 3.3. Zigzag Descriptors

The collection of $\widehat{PI}_p$s taken over all $p$ contains all the information output from our zigzag algorithm, and gives a useful overview of the model as a whole. On the other hand, they are not easily tractable statistically and are hard to interpret. We extract two descriptors from the effective persistence image, defined below.

**Births' Relative Frequency.** A useful way to summarize a persistence diagram is by counting features within a specific region of interest. In our context, it is informative to measure the rate with which new $p$-dimensional holes are created, as this reflects the model's propensity to move prompts toward each other in specific regions of space. We thus define the births' relative frequencies as

$$B_p(\ell) = \frac{\sum_{\ell_i} \omega(\ell, \ell_i) \widehat{PI}_p(\ell, \ell_i)}{\sum_{\ell_i} \omega(\ell, \ell_i) \sum_{\ell_i} \widehat{PI}_p(\ell, \ell_i)},$$
(5)

where

$$\omega(\ell, \ell_i) = |\ell - \ell_i|^\alpha$$
(6)

is a weight with varying exponent $\alpha$.[4] For negative values of $\alpha$, the average gives weight to counts with low death values, effectively tracing the fraction of short-lived features. On the other hand, positive values of $\alpha$ give more weight to long-persistent features.

**Inter-Layer Persistence.** To better track the persistence of features across layers, we can calculate the fraction of $p$-dimensional holes in one layer, $\ell_1$, that exist in another layer, $\ell_2$, as well, and have existed throughout the layers in between.[5] Mathematically it can be expressed as

$$\mathcal{Z}_p(\ell_1, \ell_2) = \frac{\sum_{\ell_1 \leq M_1, \ell_2 > M_2} \widehat{PI}_p(\ell_1, \ell_2)}{\beta_p(\ell_1)},$$
(7)

where $M_1 = \min(\ell_1, \ell_2)$; $M_2 = \max(\ell_1, \ell_2)$ and $\beta_p(\ell)$ is the Betti number, i.e. the number of alive $p$-dimensional holes at layer $\ell$.[6] We can then further summarize this quantity again by power-weighted averaging it,

$$\bar{\mathcal{Z}}_p(\ell) = \frac{\sum_{\ell_i=1}^{N_{\text{layers}}} \omega(\ell, \ell_i) \mathcal{Z}_p(\ell, \ell_i)}{\sum_{\ell_i=1}^{N_{\text{layers}}} \omega(\ell, \ell_i)}$$
(8)

where we fix one of the two layers and average over all other layers, and the weight is the same as Eq. (6). Given that the birth or death of a given $p$-dimensional hole implies the rearrangements of points in space, $\bar{\mathcal{Z}}_p$ tracks the dynamical movement of prompts' relative positions in representation space as a function of the model's depth.

## 4. Experiments

### 4.1. Models, Datasets and Benchmarks

We work with 4 models: Llama2 (Touvron et al., 2023), Llama3 (AI@Meta, 2024), Mistral (Jiang et al., 2023) and Pythia 6.9B (Biderman et al., 2023). These models are open-source decoder-only transformers, and they achieve high performance in the benchmarks we consider in this work. We analyze Llama2-7B, Llama3-8B, Mistral 7B, and Pythia 6.9B because they have 32 hidden layers and have comparable parameter sizes. In Appendix E.2 we show results for larger models as a consistency check.

The input dataset from which we take internal representations must provide a fair test of how the model processes and understands language. We consider the following datasets: 1) The Standford Sentiment Treebank (SST) dataset (Socher et al., 2013). 2) The Pile dataset (Gao et al., 2020) from which we take a subset of 10K prompts, accessible on HuggingFace.[7]3) A dataset of mathematical problems (Hendrycks et al., 2021b). 4) A dataset of codes retrieved

---

[3]Note that this operation does not modify the information about the model layers contained in the original $\text{Pers}_p(\Phi)$, as it redefines consistently all the births and deaths.

[4]This type of weighting has been used previously for topological descriptors, see e.g. (Chazal et al., 2014).

[5]We note that this is related to the generalized rank invariants in the context of multiparameter and zigzag persistence (Clause et al., 2024; Dey et al., 2024), which measures the rank of the homology maps between consecutive layers in a zigzag filtration. However, it differs in that it is normalized by the Betti number and explicitly enforces the continuity of holes throughout the intermediate layers.

[6]Note that (7) is well-defined only when $\beta_p(\ell) > 0$. If there are no $p$-dimensional holes at either $\ell_1$ or $\ell_2$, $\mathcal{Z}_p(\ell_1, \ell_2)$ should be 0 by definition. We omitted this limit case from (7) for readability.

[7]https://huggingface.co/datasets/NeelNanda/pile-10k

from GitHub.[8]

Each prompt is processed from these datasets to extract the last token at each normalization layer and the final normalization is applied to the output layer. To ensure fair comparisons and eliminate potential biases in our descriptors caused by varying point cloud sizes, all datasets are reduced to the first 8,000 prompts. Additionally, we divide the datasets into incremental subsets of $\{100, 200, ..., 1000\}$ prompts and compute the mean and standard deviation across subsets to systematically evaluate the scalability of our descriptors and to quantify their sensitivity to changes in point cloud size. For our experiments, we consider the 500 prompts subset, amounting to 16 subsets. In Appendix E.4 we perform a more detailed analysis at varying subset sizes. We use the SST dataset as a reference dataset for all the plots and show results for varying datasets in Appendix E.3.

We use 3 benchmarks for layer pruning performance evaluation: MMLU (Hendrycks et al., 2021a), HellaSwag (Zellers et al., 2019), and Winogrande (Sakaguchi et al., 2019), which have been widely used for similar purposes in previous analyses. The benchmarks are evaluated for the models with the use of the library lm-eval-harness by (Gao et al., 2024) with a 5-shot setup.

### 4.2. Zigzag persistence applied to LLM models

We generate zigzag diagrams for each model and dataset and for each homology dimension up to $p = 3$, for a range of values of $k_{NN} \in [1, 15]$. We find that 0-, 2- and 3-dimensional holes are relatively lower in number, while 1-dimensional holes reach tens of thousands of elements per layer. This behavior might be expected for a $k_{NN}$-graph-based construction since connections are dense even for low values of $k_{NN}$, especially if points are concentrated in low dimensional regions of the representation space. We examine this behavior in detail to make sure that our construction is stable for different choices on the $k_{NN}$ graph, see Appendix D for details. The choice of the hyperparameter $k_{NN}$ is done so as to maximize the total number of holes. Therefore, in what follows, we show results for our topological descriptors for 1-dimensional holes and $k_{NN} = 4$ only.

**Effective Persistence Image.** We show an example of an effective persistent image of 1-dimensional holes in Figure 2, where we use the Llama 3 8B model and the SST dataset. The x-axis represents the layer at which a 1-dimensional hole is born, and the y-axis represents persistence, i.e. death layer - birth layer. The color bar measures the amount of 1-dimensional holes on a given grid point. We see that features born after the first half of the model's depth have a higher tendency to be long-lived with respect to features

born earlier on. This aspect is going to be evident when computing also the topological descriptors, below. In the Appendix E.1, we show a wider range of images, comparing them across models by taking the element-wise difference of effective persistence images.

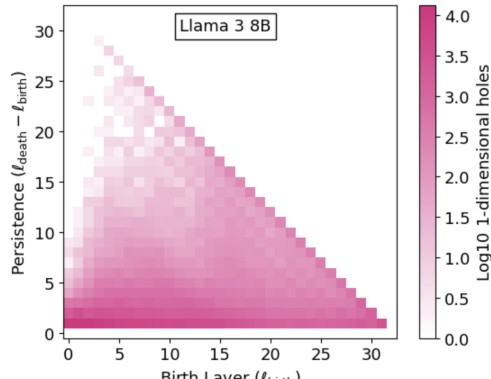

*Figure 2.* Effective persistence image of 1-dimensional holes for the Llama 3 8B model using the SST dataset, where we fix $k_{NN} = 4$. The density plot shows the number of holes (color bar) for a given birth-persistence pair (x- and y-axis), where values refer to the model layer. This plot shows that a large amount of 1-dimensional holes are short-lived and that long-lived features appear after the first half of the model.

**Births' Relative Frequency.** In the left panel of Figure 3 we show $B_1$ (Eq. 5) for Llama 3 8B on the SST dataset, for varying $\alpha = -1, 0, 0.5, 1, 2$. We can clearly distinguish two behaviors for short- and long-lived 1-dimensional holes: the former peaks at early layers and progressively decreases, while the latter peaks at middle layers. Additionally, a strong increase in births number is seen in the last few layers. We highlight a horizontal line corresponding to the uniform distribution for comparison. We complement these results in Appendix E.3 with a comparison across models and datasets, finding qualitatively similar results.

**Inter-Layer Persistence.** We show the power-weighted inter-layer persistence (Eq. (8)) in the middle and right panel of Figure 3. In the middle panel, we use Llama 3 with the SST dataset at varying $\alpha = -1, 0, 0.5, 1, 2$. As for $B_1$, we can distinguish two behaviors for short-lived and long-lived features, though now $\bar{\mathcal{Z}}_1$ traces the probability of features that are alive at a given layer to be still alive in earlier or later layers. We see that for short-lived features, this probability grows steadily until the second half of the model's depth, where it reaches a plateau, and then suddenly drops in the last few layers. On the other hand, for long-lived features, there is a peak in probability at the middle layers. We can qualitatively see the same behavior for the other models in the right panel of Figure 3, though with quantitative differences across models. We cross-check for

---

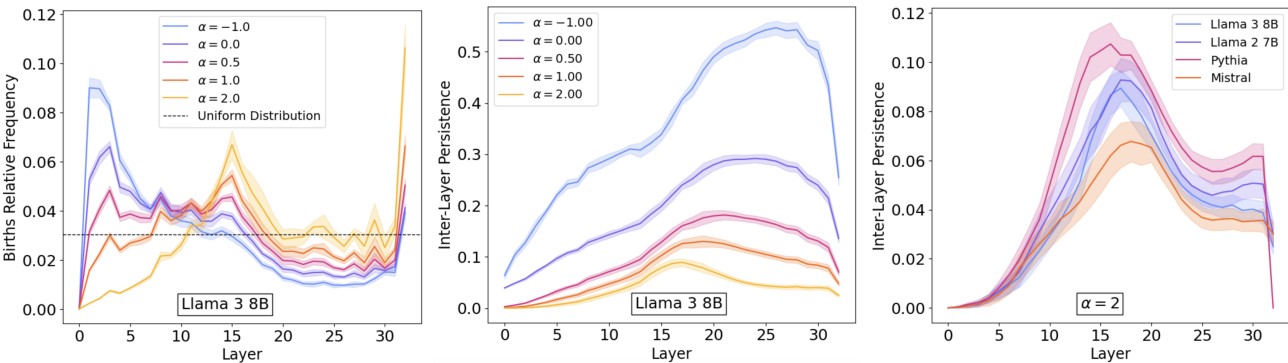

*Figure 3. Left Panel:* Births' Relative Frequency, $B_1$, as a function of model's layers for Llama 3 8B on the SST dataset, for varying $\alpha$, which traces short-(long-)lived features for negative(positive) values. Short-lived features peak early in the model and progressively decrease, while long-lived features peak in middle layers. All features experience a sharp growth in the last two layers. The dashed line represents a uniform distribution of births across the layers. *Middle Panel:* Inter-Layer Persistence as a function of model's layers for Llama 3 8B on the SST dataset, for varying $\alpha$. The persistence of short-lived features consistently grows and plateaus towards the end of the model, while long-lived features are primarily present across middle layers. *Right Panel:* Inter-Layer Persistence for Llama 3, Llama 2, Mistral and Pythia for the SST dataset at $\alpha = 2$. The models considered exhibit qualitatively similar, but quantitatively different behavior, with Pythia experiencing the higher peak, and Mistral the lowest. This seems inversely related to model's performance once pruning layers in this range. In all panels, curves and shaded regions represent the mean and standard deviation over 16 subsets of 500 prompts, respectively.

other datasets in Appendix E.3, also finding qualitatively similar, but quantitatively different results across datasets.

### 4.3. Interpretation and implications for the model's performance

In interpreting our results, it is essential to recognize that: 1) models process each token of the prompt, while we use only the last token as a proxy for the entire prompt; and 2) each prompt is processed separately from the others, such that each point moves a priori independently from the others in the representation space. Within this framework, the zigzag algorithm effectively tracks how the models dynamically organize prompts across both spatial and temporal dimensions (layers). Our findings, as illustrated in Figure 3, reveal four distinct phases:

- *Early to Middle Layers:* In the first layers, a large number of short-lived 1-dimensional holes are formed, indicating that most prompts are fastly rearranged within a few layers. This finding relates with previous work identifying local contextualization (Lad et al., 2024) and increased dimensionality (Valeriani et al., 2023).

- *Middle Layers:* In this phase, 1-dimensional holes born in middle layers have the highest probability of being long-lived than in other phases. This implies that relative positions before, but especially after these layers are kept relatively stable. This would seem related to the decreasing dimensionality found in (Cheng et al., 2024), since relevant degrees of freedom estimated by

intrinsic dimension are progressively better distinguishable and more stable to noise.

- *Middle to Late Layers:* Short-lived 1-dimensional holes born after the first layers decrease in number. At the same time, the probability that a 1-dimensional hole is short-lived increases until after the middle layers when it reaches a plateau. Concurrently, the probability (and the amount) of long-lived ones drops. This indicates a phase of relatively few short-lived adjustments in the relative positions of prompts since many of the features that formed in the middle layers are still there (because they are long-lived). We expect that these short-lived adjustments relate to a phase of specialization (Lad et al., 2024) and to a phase of relatively constant dimensionality (Valeriani et al., 2023).

- *Last Layers:* In the last two to three layers, the births' relative frequency grows rapidly while the inter-layer persistence drops. These two behaviors are compatible, given that the fraction of newborn 1-dimensional holes is large, and that there are no layers left to persist. This results suggest another strong rearrangement of points, which can be linked to the model producing the required output (Valeriani et al., 2023; Lad et al., 2024) and thus changing abruptly the position of prompts in representation space.

**Relation to model's performance.** As a test of the interpretation of the 4 phases, we perform the following exper-

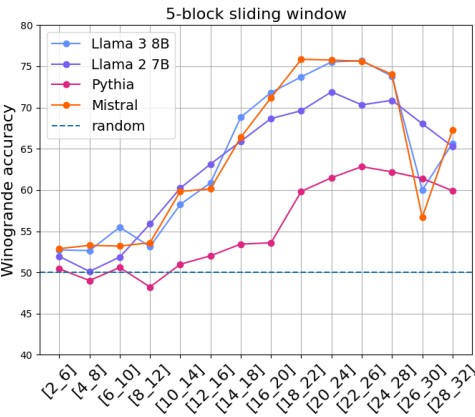

*Figure 4.* Winogrande performances for 4 models obtained with a sliding window of 5 blocks of adjacents layers and moved through the models every 2 layers. This experiment reflects 4 phases: 1) removing early layers brings performances close to random choice, 2) while performance grows to almost maximum after middle layers, and 3) plateaus; 4) removing late layers causes another drop in performance right before the end of the model.

iment: we prune blocks of layers with a sliding window from early to late layers as a way to compare the relative importance of layers in the various phases. We show the results of the experiment in Figure 4, where we show the performance of the 4 models considered for the Winogrande benchmark, as a function of the sliding window of pruned layers. We see that removing layers in the first phase significantly affects performance. After the second phase, pruning weakly affects performance, being a phase of relative adjustments. Removing the last few layers causes another drop in performance. Interestingly, the overall performance of models is inversely related to the peak height of the inter-layer persistence of long-lived features. This relation is seen also in the other limit of $\alpha < 0$. We also notice that the two best-performing models, Mistral and Llama, 3 exhibit a drop in performance when removing layers at the end of the third phase, right before the fourth. We zoom in on these layers for the MMLU benchmark where the drop is particularly evident in Appendix G.1, confirming that the drop is caused by removing the last 2-3 layers of the third phase. Importantly, all these results are qualitatively the same across the three benchmarks considered in this work.

### 4.4. Layer Pruning

Recently, measures of layer similarity have been used to identify layers that contribute minimally to the performance of LLMs. These layers can be pruned, and the performance re-evaluated to validate this assumption. Given our results in Figure 4, we can argue that layers belonging to the third phase might be pruned without affecting for model's performance. Consequently, we establish a pruning criterion

| Models | MMLU | | | HellaSwag | | | WinoGrande | | |
|---|---|---|---|---|---|---|---|---|---|
| | Full | This work | Other works | Full | This work | Other works | Full | This work | Other works |
| Llama 2 | 45.74 | 37.38 | **43.95** | 58.54 | **44.71** | 42.78 | 74.43 | **68.67** | 67.72 |
| Llama 3 | 65.07 | **53.44** | **53.44** | 61.37 | **41.60** | **41.60** | 77.10 | **70.00** | **70.00** |
| Mistral 7B | 62.40 | **53.17** | 38.20 | 62.83 | **36.67** | 34.45 | 77.35 | **66.50** | 63.76 |
| Pythia | - | - | - | 49.70 | 31.43 | **34.96** | 63.30 | 55.71 | **58.09** |

*Table 1.* **Benchmark Table.** For each benchmark, we show three columns: (i) *Full*, represents the accuracy of the model without any layer pruned. (ii) *This work*, accuracy of the model, where layers are pruned following the algorithm 2). (iii) *Other works*, accuracy obtained by considering the same amount of layer pruned estimated with our method and then computing the layer to be pruned with two different similarity measures: angular distance from (Gromov et al., 2024) and Bi-score from (Men et al., 2024). The chosen layers turn out to be the same for the two methods, so the results are condensed into one column.

based on the plateau observed in the inter-layer persistence of short-lived features. Specifically, we prune layers that lie within $10\%$ of the maximum value of $\bar{\mathcal{Z}}_1$. This is computed for each different model, using the Pile dataset as proxy.[9] We show a schematic summary of the algorithm in Appendix G.2.

We compare our layer pruning methods to recent work (Gromov et al., 2024) and (Men et al., 2024) performing pruning using similarity measures. Both approaches are designed to take as input the desired number of layers to prune $N_{\mathrm{prune}}$ and measure performance as $N_{\mathrm{prune}}$ grows. For a fair comparison, we feed the number of layers cut by our method as an input to the other two methods, and verify which layers they select to cut given this input, and the corresponding performance. We show which layers are cut for each method in Table 2 in Appendix G.2. Interestingly, both considered methods from (Gromov et al., 2024) and (Men et al., 2024) give the same result at fixed $N_{\mathrm{prune}}$, thus we refer to them simply as "other works". We show performance results in Table 1,[10] where in bold we indicate the layer pruning method that has better or equal performance with respect to the other method. Despite often selecting different layers, our zigzag-based pruning strategy achieves comparable results to methods from (Gromov et al., 2024) and (Men et al., 2024).

## 5. Conclusions

Recent work has argued in different ways that large language models process inputs across layers through distinct phases,

---

[9]We use Pile since it is characterized by a broad range of topics, representing a wider range of prompts. Note that the shape of the curve around the peak of $\bar{\mathcal{Z}}_1$ is approximately similar across datasets (see Figure 9 in Appendix).

[10]Results for Pythia on MMLU tasks are not shown because the model is not designed for following the format of the tasks, as shown in (Biderman et al., 2023).

and that understanding these phases is important for the models' interpretation. We exploit topological data analysis tools to build descriptors that allow to statistically characterize the dynamics of prompts within internal representations of large language models. Based on this characterization, we distinguish four phases and connect them to the model's behavior through experiments based on layer pruning and performance benchmarking. Our method consistently provides qualitatively similar results across different models, datasets, and parameter selections. Simultaneously, our topological descriptors allow for quantitative differentiation across models and datasets, creating opportunities for experiments designed to address more specific and practical questions regarding particular models or datasets.

There are several limitations in our study that future research could address. First, while our method shows robustness across hyperparameters within the framework, these choices need not be optimal. Defining an appropriate criterion for connecting points in the representation space, and consequently, a filtration, is a delicate task in TDA that could require further investigations to detail the impact of the various choices on the construction of the filtration. Secondly, our study primarily focuses on static, pre-trained models. Extending this framework to track the evolution of internal representations during training might provide important insights into model efficiency and behavior.

## 6. Reproducibility

All the results contained in this work are reproducible by means of a GitHub repository that can be found at this link https://github.com/RitAreaSciencePark/ZigZagLLMs.

## Impact Statement

This paper presents work whose goal is to advance the field of Machine Learning. There are many potential societal consequences of our work, none which we feel must be specifically highlighted here.

## Acknowledgments

We thank Mathieu Carrière and Magnus Botnan for helpful discussions on the TDA implementation and suggestions on the zigzag algorithm. M.B. Y.G. and G.P. are partially supported by the Programma Nazionale della Ricerca (PNR) grant J95F21002830001 with title "FAIR-by-design". K.V. was partially supported by Programma Nazionale della Ricerca (PNR) grant J95F21002830001 with the title "FAIR-by-design" during his visit to Area Science Park while this project was in its development phase. A.A. and A.C. were supported by the project "Supporto alla diagnosi di malattie rare tramite l'intelligenza artificiale" CUP: F53C22001770002. A.A. and A. C. were supported by the European Union – NextGenerationEU within the project PNRR "PRP@CERIC" IR0000028 - Mission 4 Component 2 Investment 3.1 Action 3.1.1.

We thank Area Science Park supercomputing platform OR-FEO made available for conducting the research reported in this paper and the technical support of the Laboratory of Data Engineering staff.

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

# A. Mathematical Formulation of Zig Zag Persistence

Zigzag persistence is a computational topology method that extends classical persistent homology to handle more complex data structures and filtration processes. Unlike standard persistence, which analyzes a single sequence of spaces filtered by inclusion, zigzag persistence allows for the exploration of data where sequences of spaces and maps can move both forward and backward.

A *zigzag filtration* of topological spaces is a sequence:

$$\chi\colon \mathbb{X}_1 \longleftrightarrow \mathbb{X}_2 \longleftrightarrow \cdots \longleftrightarrow \mathbb{X}_n, \qquad (9)$$

where each $\mathbb{X}_i$ is a topological space and each arrow $\longleftrightarrow$ represents a continuous function pointing forwards $\mathbb{X}_i \longrightarrow \mathbb{X}_{i+1}$ or backwards $\mathbb{X}_i \longleftarrow \mathbb{X}_{i+1}$.

If we apply a homology functor $H_p$ with coefficients in a field $\mathbf{k}$ to such a filtration, we get a zigzag filtration of $\mathbf{k}$-vector spaces, called *zigzag module*:

$$H_p(\chi)\colon H_p(\mathbb{X}_1) \longleftrightarrow H_p(\mathbb{X}_2) \longleftrightarrow \cdots \longleftrightarrow H_p(\mathbb{X}_n). \qquad (10)$$

It is proven in (Carlsson & de Silva, 2010) that the algebraic classification of zigzag modules resembles Gabriel's classification of the persistence module described in (Gabriel, 1972). In particular, every finite-dimensional zigzag module, i.e. for which all the $\mathbf{k}$-vector spaces in the sequence that are finite-dimensional, can be decomposed as a direct sum of interval modules, where a (finitely indexed) *interval module* is a module of the form:

$$\mathcal{I}_{[b,d]}\colon I_1 \longleftrightarrow I_2 \longleftrightarrow \cdots \longleftrightarrow I_n, \qquad (11)$$

where $I_i = \mathbf{k}$ for $b \le i \le d$, and $I_i = 0$ otherwise, and every arrow of the form $\mathbf{k} \longleftarrow \mathbf{k}$ or $\mathbf{k} \longrightarrow \mathbf{k}$ is the identity map. Moreover, the list of summands is unique up to reordering.

The *zigzag persistence diagram* of a filtration $\chi$ in dimension $p$ is the multiset of intervals $[b,d]$ corresponding to the list of interval summands $\mathcal{I}_{[b,d]}$ of $H_p(\chi)$. In other words,

$$\mathrm{Pers}_p(\chi) = \{[b_j, d_j]\colon j \in J\} \Longleftrightarrow H_p(\chi) \cong \bigoplus_{j \in J} \mathcal{I}_{[b_j, d_j]} \qquad (12)$$

Each interval $[b,d]$ is called *persistence interval* and is thought of as a persistent homological feature of $\chi$ that appears at time $b$ (referred to as the "birth") and disappears at time $d$ (referred to as the "death[11]").

---

[11]In our setting we say a $p$-dimensional holes "dies", we mean that the corresponding homology class no longer persists in subsequent layers. In the zigzag filtration, this happens when the hole is no longer represented by an independent equivalence class in the homology group.

In our approach, the use of intersection layers is essential for computing zigzag persistence, as it allows the construction of injective maps between the $k_{\mathrm{NN}}$ complexes of model layers (see (2))[12]. Since our primary goal is to analyze the topological changes between model layers, we eliminate the construction of intersection layers while preserving the topological features by shifting each persistence interval such that the birth and death times occur strictly within the layers.

For an interval $[b,d]$ in the zigzag persistence diagram of dimension $p$ of filtration 2, the mapping that enables a bijective transformation to a new interval $[\hat{b}, \hat{d}]$[13] only across model layers is defined as follows:

$$\hat{b} = \begin{cases} b+1 & \text{if } b \text{ is an intersection layer} \\ b & \text{otherwise} \end{cases},$$

$$\hat{d} = \begin{cases} d+1 & \text{if } d \text{ is an intersection layer} \\ d & \text{otherwise} \end{cases} \qquad (13)$$

The relationship between the persistence image and the effective persistence image for $p$-dimensional holes, denoted respectively by $PI_p$ and $\widehat{PI}_p$, where $b, d$ are the model layers indexed by even numbers, is described by the following system of equations:

$$\begin{cases} \widehat{PI}_p(0,0) = & PI_p(0,0) \\ \widehat{PI}_p(b/2, d/2) = & PI_p(b,d) + PI_p(b-1,d) \\ & + PI_p(b, d-1) + PI_p(b-1, d-1) \\ \widehat{PI}_p(b/2, \infty) = & PI_p(b, \infty) + PI_p(b-1, \infty). \end{cases}$$

# B. Zigzag algorithm

The zigzag algorithm is schematically described below.

It exploits two existing public codes that were developed for zigzag computations: DIONYSUS2 (Morozov) and FASTZIGZAG (Dey & Hou, 2022). DIONYSUS2 is a C++ library for computing persistent homology, with a specific library for zigzag persistence. In our case, it has the role of extracting the filtration $f$ and computing the $times$ array, i.e. the list of layer indices to be associated with the birth and death of features. FASTZIGZAG allows to calculate efficiently the persistence diagram $\mathrm{Pers}_p(\Phi)$ by converting the input zigzag filtration to a non-zigzag filtration of an equivalent complex with the same length, and it then converts the

---

[12]An alternative method for constructing these maps and obtaining the zigzag persistence diagram is to use a filtration where, instead of intersections, the union of the complexes from two consecutive layers is considered. However, the Diamond Lemma, as discussed in (Carlsson et al., 2009), guarantees that both the intersection- and union-based filtrations encode the same homological information.

[13]By construction, all resulting intervals contain even numbers, as the model layers are indexed with these numbers.

**Algorithm 1** Zigzag algorithm

**Require:** $model, dataset, k_{NN}, m$
  $reps \leftarrow$ extractRepresentations$(model, dataset)$
  $K \leftarrow []$
  **for** $i \leftarrow 1$ to $model$.getNumLayers() **do**
    $graph \leftarrow$ kNearestNeighborsGraph$(reps[i], k_{NN})$
    $K$.append(graphExpansion$(graph, m)$)
  **end for**
  $K_{int} \leftarrow$ computeIntersectionLayers$(K)$
  $f, times \leftarrow$ computeFiltrationTimes$(K, K_{int})$
  $\Phi \leftarrow$ FastZigZag$(f, times)$

obtained persistence intervals back to zigzag. The computational cost of our algorithm is $\mathcal{O}(n^2 * N_{layers}) + \mathcal{O}(m^\omega)$ where the first part is the $K_{NN}$ graph creation cost for the input dataset at each layer, and the second part is the theoretical cost of FastZigZag with $\omega < 2.37286$. The algorithm performs well even for the relatively large datasets we employ for this analysis: with 10K points embedded in a space with dimension $d = 4096$, a number of neighbors for the $k_{NN}$ graph of $k_{NN} = 10$, and a maximum homology dimension of $m = 10$ on an AMD EPYC 7H12 it takes approximately 2 hours.

## C. A Toy Example to Visualize Zigzag Barcodes

To demonstrate the zigzag persistence visualization framework, we analyze a simple calendar arithmetic task on the Mistral model. We use prompts from the form

"Let's do some calendar math. Four months from
**[MONTH] is**"

where [MONTH] cycles through all twelve months. This example was used in (Engels et al.) to analyze circular representations in language models.

We extract hidden representations from all 33 transformer layers[14] and apply zigzag persistent homology with $k = 2$ neighbors and maximum simplex dimension 3. The point cloud at each layer comprises 12 tokens corresponding to the 12 different month prompts. We analyze two token positions: month tokens (semantic input) and answer tokens (given by the **is** token). So the point cloud at each layer comprises of 12 tokens for each prompt

Figure 5 shows the analysis for both token types. The persistence barcode for month tokens (top panel) exhibits a prominent feature from layers 5-14, while answer tokens (bottom panel) show persistent structure emerging later (layers 21-32).

---

[14]Layer 0 is the embedding layer.

The visualization demonstrates two key capabilities of zigzag barcodes:

- **Layer-wise tracking:** Barcodes reveal when topological features emerge and disappear across network depth.

- **Differential patterns:** Different token types exhibit distinct persistence signatures.

This toy example illustrates the framework's ability to visualize geometric structure evolution in transformer representations.

## D. Combining the $k_{NN}$ graph with the Vietoris-Rips complex

The k-Nearest Neighbors ($k_{NN}$) complex is built by expanding the corresponding $k_{NN}$ graph to a fixed dimension $m$. A key limitation of the $k_{NN}$ complex is that it ranks points by proximity without considering their actual distances. As a result, once $k$ is fixed on each layer, each point is connected to its k-Nearest Neighbors, regardless of the absolute distances involved. In our setting, the number of connected components (the Betti [15] number $\beta_0$) of the $k_{NN}$ complexes as a function of the layers tends to be unity, i.e. the whole complex is connected, even for relatively small values of $k_{NN} \gtrsim 6$. This implies that connected components contain no useful topological information on the internal representations.

To address this issue, we follow the approach in (Naitzat et al., 2020), which combines the $k_{NN}$ complex with the Vietoris-Rips complex. Starting from the $k_{NN}$ graph, the idea is to introduce a threshold radius $R$ on each layer and use it to filter out edges of the graph whose lengths are less than or equal to $R$, and then expand, denoting this new complex $k_{NN}$-VR. This filtering step allows us to focus on longer-range connections, uncovering significant topological features that may be hidden by shorter, more local connections.

To ensure consistency across layers, we select the radius $R$ in each layer such that the number of connected components, $\beta_0$, of the $k_{NN}$ complex falls in a pre-determined range. We then compute the observables presented in this work and verify the results. For clarity, we refer to $k_{NN}$ complex the construction used in the main body, and $k_{NN}$-VR complexes the one presented in this section. For the sake of conciseness, we present only results for the inter-layer persistence $\bar{\mathcal{Z}}$.

In Figure 6 we show the inter-layer persistence of 1-

---

[15]Betti numbers have been used in previous works (Naitzat et al., 2020; Suresh et al., 2023) for interpreting internal representations of neural networks. However, they describe each layer independently from the others, which is not the purpose of this work.

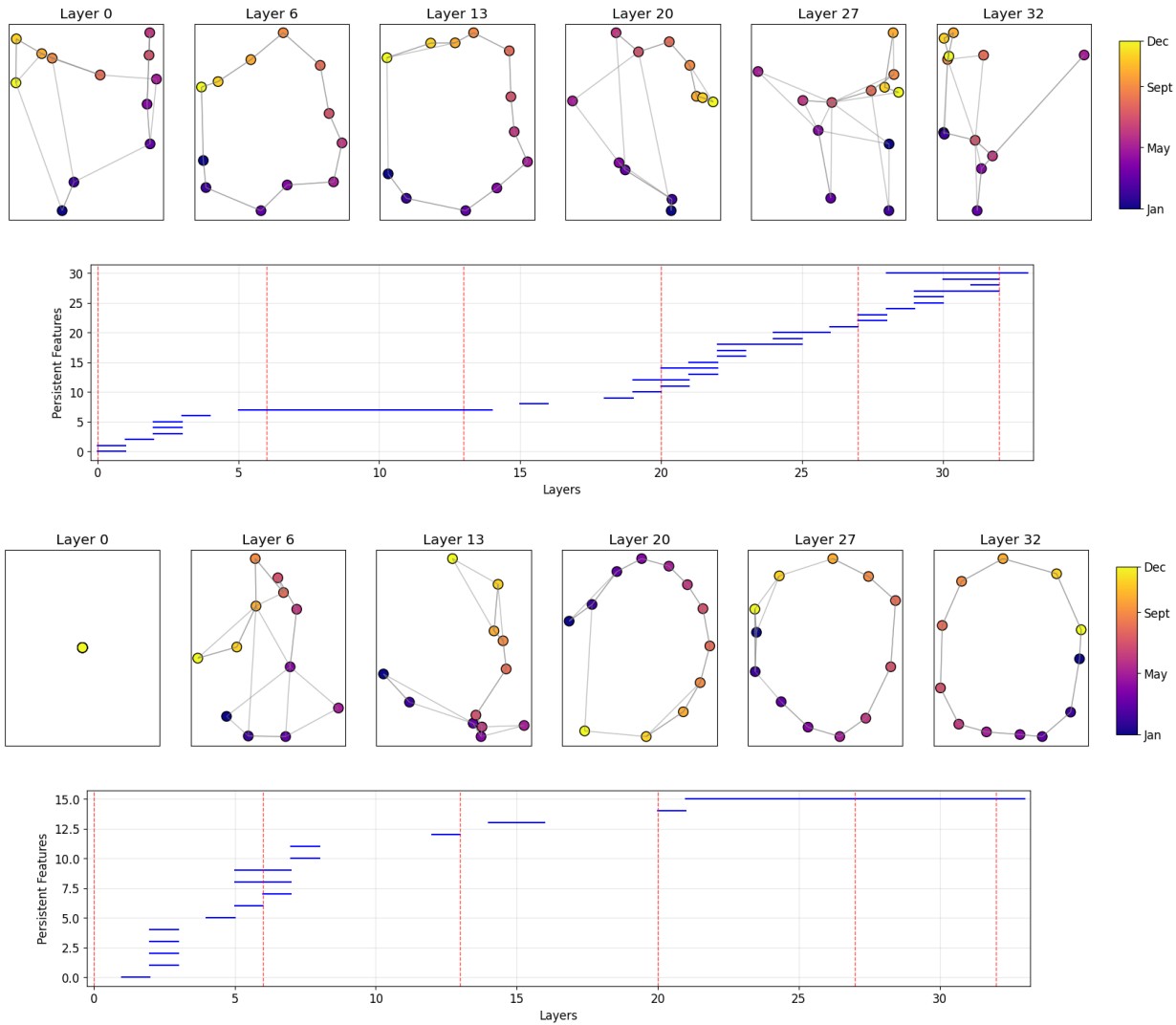

*Figure 5.* Zigzag persistence analysis of calendar arithmetic tokens. *Top panel:* Month token representations exhibit persistent topological structure emerging at layer 5 and persisting until layer 14. The points are plotted with the first and second component of PCA, respectively X and Y axis. *Bottom panel:* Answer token representations show late-emerging persistent structure from layers 21-32, with the points plotted on the first and second component of PCA. For each panel, the upper half displays $k$-nearest neighbor graphs ($k = 2$) for the 12 prompts across selected transformer layers, while the lower half shows the corresponding 1-dimensional persistence barcodes tracking topological feature evolution across all 33 layers. The red dotted lines in the barcode plots indicate the specific layers visualized in the upper half.

dimensional holes of the $k_{\mathrm{NN}}$ and the $k_{\mathrm{NN}}$-VR complexes and the 0-dimensional holes of the $k_{\mathrm{NN}}$-VR complexes computed by imposing $\beta_0 = 500 \pm 100$. [16] We observe all three curves are qualitatively similar. This indicates the stability of the results, even when removing a considerable amount of short edges. Moreover, we observe the same behavior also on 0-dimensional holes, now that we modified the complex such that their statistics are large enough to reliably compute persistence. We argue this indicates a universal (in

homology) tendency to retain relational connections among particles in the middle-late layers of the model.

## E. Consistency of Results

### E.1. Effective Persistent Images across Models

Given a fixed dataset, effective persistent images can be calculated across models and subtracted element-wise to highlight differences in how the models process the same information. We show all the possible comparisons for the 4 models considered in this work in Figure 7, which we

---

[16]We checked that results are stable as long as $\beta_0$ is much lower than the total number of points.

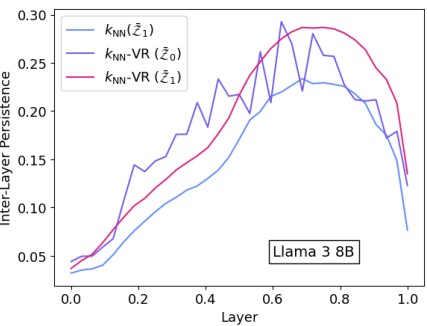

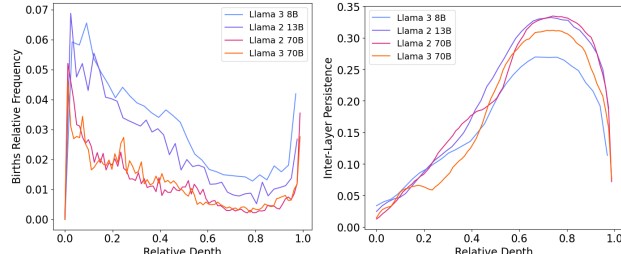

*Figure 8.* Births' relative frequency (left) and inter-layer persistence (right) as a function of the models' depth for larger models, namely Llama 2 13B, Llama 2 70B, and Llama 3 70B, compared to Llama 3 8B, computed for the SST dataset with weight $\alpha = 0$.

*Figure 6.* Inter-Layer Persistence with weight $\alpha = 0$ as a function of model layers computed for Llama3 8B on the SST dataset for both $k_{NN}$ and $k_{NN}$-VR complexes. We impose the number of connected components, $\beta_0 = 500 \pm 100$ to build the $k_{NN}$-VR complexes.

Llama 3 8B model and choose $\alpha = 0$ as weight for both the births' relative frequency and inter-layer persistence. We show results averaged over subsets of size 500 points in Figure 9. We note qualitative similarity for both descrip-

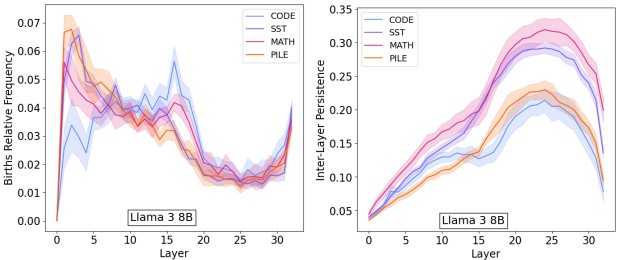

*Figure 9.* Births' Relative Frequency and Inter-Layer Persistence for weight $\alpha = -0$ as a function of model layers for Llama 3 8B for a range of datasets, averaged over 16 subsets of size 500.

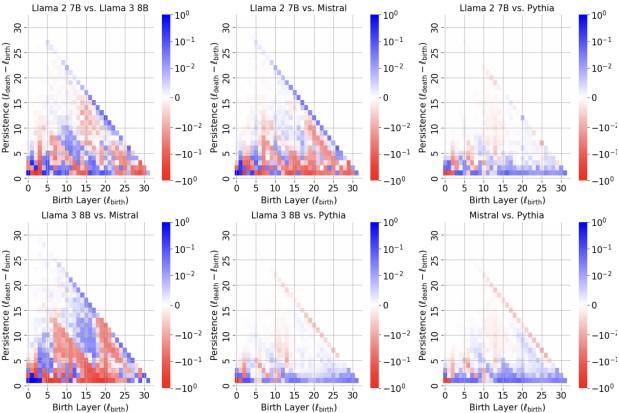

*Figure 7.* Element-wise difference of effective persistence image calculated for Llama 2, Llama 3, Mistral and Pythia on the SST dataset. The color bar indicates a normalized difference between $-1$ and $1$, on a logarithmic scale.

calculate for the SST dataset. We can observe clear patterns in differences across models, reflecting what is observed in Figure 3.

### E.2. Larger Models

We verify that our topological descriptors exhibit the same qualitative results for larger models, namely Llama 2 13B, Llama 2 70B and Llama 3 70B, using the SST dataset. We show the births' relative frequency and inter-layer persistence in Figure 8 in the left and right panels, respectively. As a representative value, we choose a weight of $\alpha = 0$ for both descriptors, which gives equal weight to short- and long-lived features.

### E.3. Varying Datasets

We test our topological descriptors on 4 different datasets, as presented in Section 4. As a reference, we consider the

tors across all datasets, though quantitative differences can clearly be seen especially for inter-layer persistence. Interestingly, we observe that the code dataset has a slight divergent behaviour in middle layers. To investigate this further, we filter the Code dataset for 5 programming languages with different levels of verbosity (C, Java, HTML, Markdown, Python) and for each one we extract 10K prompts. We then calculate the inter-layer persistence for $\alpha = -1$ and $\alpha = 2$ for these datasets so as to highlight separately short- and long-lived features. In this case, we average over 8 subsets of size 1000 for computational convenience. We show results in Figure 10. From these calculations, we gather that most programming languages generally exhibit a drop in the amount of short-lived features in the middle layer, the effect being more visible for verbose codes (java, C). This feature is not seen for other types of datasets. These results deserve further investigation, which we leave to future work.

### E.4. Results as a Function of Varying Subsets

Here we show that our topological descriptors show consistent results across various subset sizes. While the variance

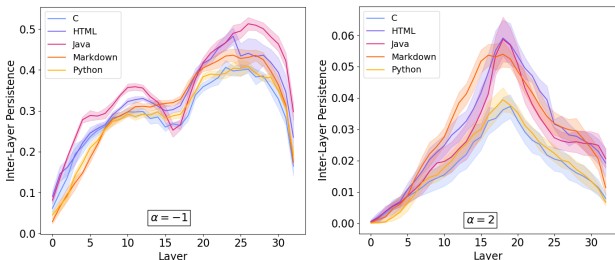

*Figure 10.* Inter-Layer Persistence for weights $\alpha = -1$ (left panel) and $\alpha = 2$ (right panel) as a function of model layers for Llama 3 8B for a range of programming languages, averaged over 8 subsets of size 1000.

increases for smaller subsets, descriptors computer over different subset sizes are within a standard deviation. We show these results for both the births' relative frequency and the inter-layer persistence calculated for the Llama 3 8B model on the SST dataset in Figure 11.

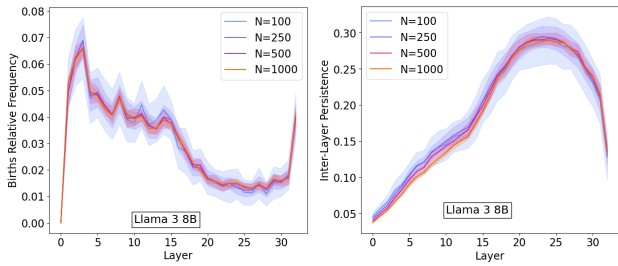

*Figure 11.* Births' relative frequency (left) and inter-layer persistence (right) with weight $\alpha = 0$ for the Llama 3 8B model computed on the SST dataset for different subset sizes as a function of model layers.

**Variance of $\bar{\mathcal{Z}}_1$ as a function of data points.** Given all the subsets with $\{100, 200, ..., 1000\}$ points, we can calculate how the variance of these subsets scales as compared to the size of the subset. As a test case, we take the Llama 3 8B model with the SST dataset and compute the inter-layer persistence at weight $\alpha = 0$ over all the subsets. We then plot the variance of each subset as a function of the subset size for different layers. We choose 4 layers so that they are roughly representative of the dynamical phases identified in the main text. We show results in Figure 12, where we overlay a fitted curve in black. Apart for the first layers, where the variance grows with the number of points, N, in later layers the relation seems to be approximately $\sigma^2 \propto N^{3/2}$.

### E.5. Additional result across models

In this section we present supplementary evidence of the consistency of the results across models for our descriptors,

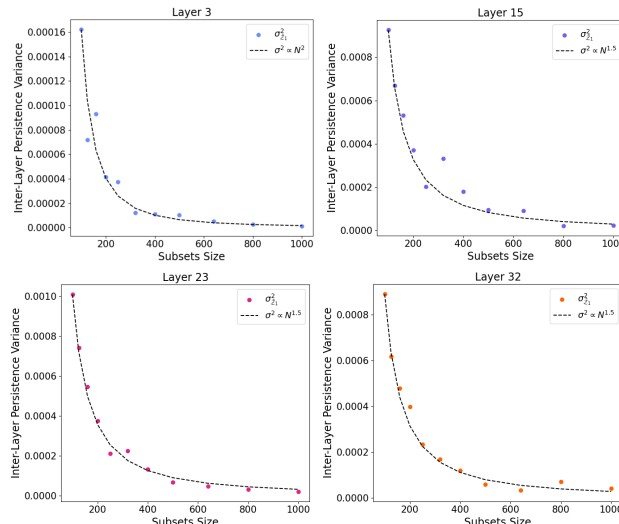

*Figure 12.* Variance of the inter-layer persistence with weight $\alpha = 0$ for the Llama 3 8B model computed on the SST dataset as a function of subset sizes. We show four different layers: 3, 15, 23 and 32. The black dashed lines represent a fitting function.

effective persistence, Births' Relative Frequency and Inter-Layer Persistence on Llama 2, Mistral, and Pythia, in Figure 13.

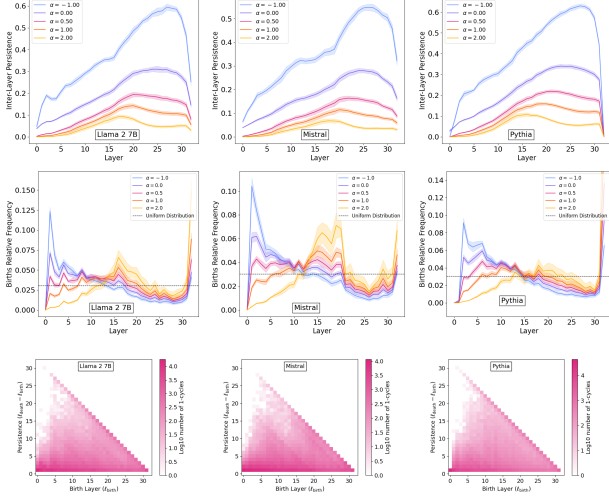

*Figure 13.* Supplementary plots for Llama 2, Mistral, and Pythia on the SST dataset. The first row displays Inter-layer persistence, the second row shows the Births' relative frequency, and the third row presents the effective persistent images.

## F. A shuffling test

As a test of our topological descriptors and the phases seen in Section 4, we perform a shuffling of tokens within the prompts of the SST dataset, as a way of destroying the structure and semantic coherence of the prompts, without

modifying their unigram frequency distribution (see e.g. (Cheng et al., 2024) for an application of shuffling to internal representations of transformers).

In Figures 14 and 15, we show the births' relative frequency and the inter-layer persistence for shuffled and structured prompts. For the former, we can see a clear difference in behavior on the birth of the long-lived features between the shuffled and structured cases, the peak at middle layers being higher for shuffled prompts. A reversed trend is seen for the inter-layer persistence. Overall, the frequency of births of short-lived features is not significantly affected by the shuffling, while the inter-layer persistence drops on the second half of the model's depth. These findings deserve further investigations, which we postpone to future work.

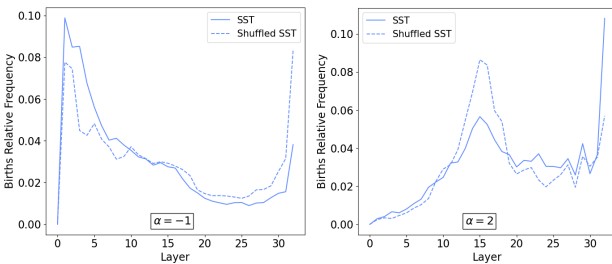

*Figure 14.* Births' relative frequency at weight $\alpha = -1$ (left) and $\alpha = 2$ (right) as a function of model layers for the Llama 3 8B for shuffled and unshuffled prompts from the SST dataset.

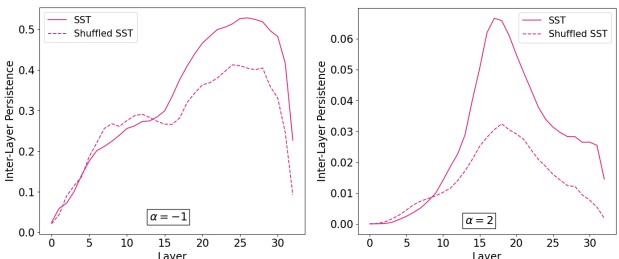

*Figure 15.* Inter-layer persistence at weight $\alpha = -1$ (left) and $\alpha = 2$ (right) as a function of model layers for the Llama 3 8B for shuffled and unshuffled prompts from the SST dataset.

# G. More on Pruning

## G.1. Sliding Window on Other Benchmarks

We can test the sliding window experiments on the other two benchmarks, MMLU and Hellaswag, show in Figures 17 and 16, respectively.

For the case of MMLU, we zoom in on the drop in performance seen at the end of the third phase: we show the performance of the MMLU benchmark against block sizes of 5, 3, and 2 adjacent layers with sliding windows of 2, 1

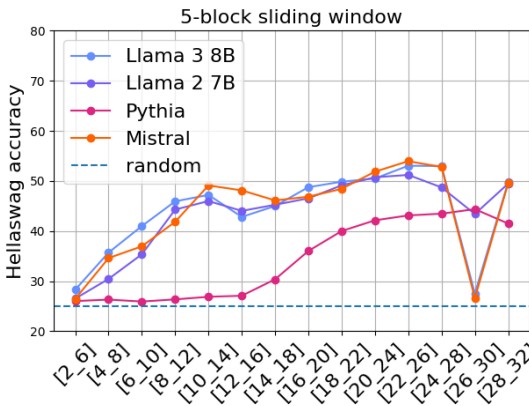

*Figure 16.* Hellaswag 5-shot benchmark run on Llama3 8B, Llama2 7B, Mistral and Pythia. A sliding window of size 5 is applied to cut blocks every 2 layers.

and 1 for the left, middle and right panels, respectively. We see that performance is at the level of random choice during the increasing phase and it maximizes close to the maximum Inter-Layer Persistence during the plateau phase. Consistently with the Winogrande benchmark, we see a drop in performance right in correspondence with the decreasing phase. For both Llama and Mistral, the relevant layers are a few layers before the last. This finding deserves a closer investigation, which we leave for future work.

## G.2. Layer pruning algorithm

Here we schematically describe the algorithm for layer pruning used to produce results presented in Table 1.

---

**Algorithm 2** Pruning algorithm

---

**Require:** $\bar{\mathcal{Z}}_1, model, threshold,$
  $max \leftarrow \max(\bar{\mathcal{Z}}_1)$
  $layersToRemove \leftarrow []$
  **for** $l \leftarrow 1$ to $model.getNumLayers()$ **do**
    **if** $\bar{\mathcal{Z}}_1[l] > max * threshold$ **then**
      $layersToRemove.append(l)$
    **end if**
  **end for**
  $model.removeLayers(layersToRemove)$

---

| Models | $N_{\mathrm{prune}}$ | Our method | Other works |
|--------|------|------------|-------------|
| Llama 2 | 8 | [20-27] | [21-28] |
| Llama 3 | 9 | [20-28] | [20-28] |
| Mistral | 10 | [20-29] | [19-28] |
| Pythia | 11 | [19-29] | [18-28] |

*Table 2.* Table with the list of models and the layers that we cut for the pruning experiment. With a given $N_{\mathrm{prune}}$ we show the layers cut with our method and for the Angular Distance and the Bi Score (Other works).

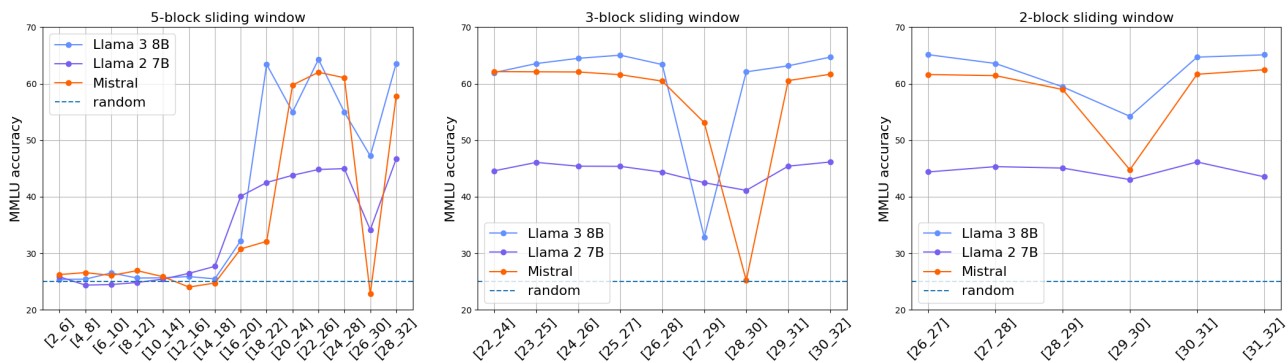

*Figure 17.* MMLU 5-shot benchmark run on Llama3 8B, Llama2 7B and Mistral. The different benchmarks shown are done by cutting blocks of layers with a fixed size and by changing the starting point with a sliding window. *Left Panel*: benchmark made with a block size of 5 and sliding windows of 2, *Middle Panel*: benchmark made with block size of 3 and sliding windows of 1, *Right Panel*: with a block size of 2 and sliding window of 1.

| Model | CMMLU | | | Commonsense-QA | | | WSC | | |
|---|---|---|---|---|---|---|---|---|---|
| | full | this work | other works | full | this work | other works | full | this work | other works |
| llama 2 | 32.17 | 28.51 | **30.00** | 55.94 | 38.40 | **52.83** | 88.63 | **84.60** | 75.80 |
| llama 3 | 50.96 | 34.02 | 34.02 | 73.45 | 65.93 | 65.93 | 85.70 | 80.94 | 80.94 |
| Mistral | 44.47 | **38.84** | 29.68 | 69.78 | **62.33** | 30.62 | 87.18 | 69.60 | **72.15** |
| Pythia | - | - | - | - | - | - | 81.67 | 60.06 | **71.43** |

*Table 3.* Supplementary 5-shots benchmarks done with the same methodology as in Table 1.

## G.3. Additional benchmarks

In Table 3, we present additional benchmarks beyond those presented in the main text, using the pruning method outlined in 4.4. Results are overall consistent with what find with benchmarks in the main text.

