# OpenReview forum: "Persistent Topological Features in Large Language Models"
_ICML.cc/2025/Conference — ICML 2025 poster_

### Official Review · Reviewer_yjVt · 2025-03-11

**Overall Recommendation:** 4

**Summary:**

The authors of this paper explore the applicability of Zigzag filtrations in the Persistent Homology framework for feature extraction in LLM analysis. They propose building filtration on top of simplicial complexes defined by kNN-neighborhoods instead of proximity-induced cliques more commonly used in the persistent homology framework.

Authors use those filtrations to analyze the evolution of internal representations of a set of texts across different layers of LLM (a text is represented by its last token; thus, a set of texts is a point cloud). They introduce a metric of similarity between layers topology and identify layers that contribute little to the model performance. Finally, this paper covers a method of model pruning by removing blocks of the identified layers.

## update after rebuttal

I think that with the proposed changes, this paper will be an interesting contribution to the research area, although some concerns remain regarding the advantages of the proposed method compared to the "usual" simplicial filtrations. I have raised my score to 'Accept'.

**Claims And Evidence:**

Claim of “Identification of Phases of Prompt processing” is too broad - prompts can’t be truly represented with only one token (as it is done in the paper).

**Essential References Not Discussed:**

I didn’t find any critically important references that were not covered in the literature review.

**Experimental Designs Or Analyses:**

1)  Why was used sliding window of 5, why no other sizes were reported?

2) Is there any difference (in terms of identified 4 phases) for encoder-only models (in the paper only decoder models are covered)?

**Methods And Evaluation Criteria:**

Seems reasonable.

**Other Comments Or Suggestions:**

1) Only one task of layer pruning was explored. Some other tasks, especially those focusing on topology of individual texts, would be much appreciated.

**Other Strengths And Weaknesses:**

Strengths:

S.1. Applications of zigzag filtrations had little coverage in the literature as of now.

S.2. The paper is well-written and easy to follow.

Weaknesses:

W.1. The title feels a little too broad. Different methods of extraction of persistent features from the internal representations of LLMs was covered in lots of previous works, while this paper introduces only one type of new filtration (new features) and explores only one downstream task.

**Questions For Authors:**

1) Have you performed your main experiments only with 0-, and 1-dimensional topological features?

2) From what dimensionality topology of $kNN$-complexes becomes trivial?

3) Could you please provide for Figures 4 plot of average length of intervals in (classic) persistence diagrams for each of the layers for comparison?

**Relation To Broader Scientific Literature:**

Application of the already existing topological method to a new problem.

**Theoretical Claims:**

In this paper no theorems or statements requiring mathematical proofs were made.

---

> ### Author Rebuttal · Authors · 2025-03-31
>
> We thank the reviewer for their useful feedback. They recognize the novel approach of using zigzag persistence in the context of interpretability of NNs and the clarity of presentation. Additionally, they raise a few points which we address element-wise below:
>
> > Claim of “Identification of Phases of Prompt processing” is too broad - prompts can’t be truly represented with only one token (as it is done in the paper).
>
> While we agree with the referee that an analysis of whole sentences would allow for finer-grained information about phases, we chose to simplify the analysis computationally by adopting the common practice in studies of the geometry of representations of looking at the last token only.
>
> > Why was used sliding window of 5, why no other sizes were reported?
>
> Figure 15 shows the analysis for different sliding windows, which we use to zoom in on a phase in which accuracy drops significantly. In other ranges, the size of the window would not change results significantly, if kept small enough.
>
> > Is there any difference (in terms of identified 4 phases) for encoder-only models (in the paper only decoder models are covered)?
>
> We believe that a difference might indeed be seen for encoder-only models. For those models, considering the last token as a proxy for the whole sentence would not be correct. Along these lines, previous studies of protein language models (see e.g. Valeriani et al. 2023) have considered averaging over tokens. We deserve these studies to future work.
>
> > Only one task of layer pruning was explored. Some other tasks, especially those focusing on topology of individual texts, would be much appreciated.
>
> We thank the referee for suggesting exploring different downstream tasks. Indeed previous work has considered using geometric quantities (e.g. intrinsic dimension) to identify human text from AI-generated text (Tulchinski et al 2024). An analysis of individual sentences would imply using our algorithm on the tokens of a prompt, i.e. without reducing the whole prompt to a single prompt. Such analysis would be interesting, though quite different from the present work since the manifolds in the two cases would represent rather different dynamics.
>
> > Have you performed your main experiments only with 0-, and 1-dimensional topological features?
>
> We have performed our calculations for up to 3-dimensional holes. However, as mentioned in section 4.2, 0-,2- and 3-dimensional holes have relatively low number counts, making it hard to draw solid conclusions from them. A brief discussion on which homology dimension contains relevant information is included in Appendix C.
>
> >From what dimensionality topology of kNN-complexes becomes trivial?
>
> We find non-zero counts up to 3-dimensional holes. This is specific to kNN complexes which are typically more connected as compared to other complex such as e.g. Vietoris-Rips complexes.
>
>
> > Could you please provide for Figures 4 plot of average length of intervals in (classic) persistence diagrams for each of the layers for comparison?
>
> We apologize with the referee, but we did not understand this question. Are they asking to perform persistent homology at each layer for comparison?

---

> > ### Comment · Reviewer_yjVt · 2025-04-09
> >
> > Thank you for your responses.
> >
> > > We apologize with the referee, but we did not understand this question. Are they asking to perform persistent homology at each layer for comparison?
> >
> > I apologize for the unclear question formulation.
> > I was asking about the average length of lifespan intervals for simplicial filtration (Vietoris-Rips) for each of the LLM layers. I was interested in  its comparison to the number of intervals (from Zigzag filtration) that "pass" through each layer (Figure 4). For me, it would be very interesting to understand whether it is possible to achieve similar results using tools based on simplicial filtrations.
> >
> > I have read other reviews and responses to them. I think that with the proposed changes, this paper will be an interesting contribution to the research area, although some concerns remain regarding the advantages of the proposed method compared to the "usual" simplicial filtrations. I am willing to raise my score to 'Accept'.

---

### Official Review · Reviewer_N5pr · 2025-03-13

**Overall Recommendation:** 3

**Summary:**

The paper tackles the problem of understanding how LLMs work by looking at how layers sequentially transform prompts. Unlike current art that only provides static views of internal representations, the paper uses zig-zag persistence across layers obtained from simplicial complexes built using kNN. Based on their empirical analysis, the authors identify 4 phases of LLM processing. Finally, the paper shows how we can leverage insights from the initial analysis to perform layer pruning, obtaining competitive results with SOTA.

---

**Post-rebuttal update**

I want to thank the authors for their efforts to address my concerns, including the additional plots/experiments. After reading the other reviews/responses, I decided to keep my initial score.

**Claims And Evidence:**

The paper claims a "fast and scalable pipeline to characterize Trasformer's layers". In particular, the paper establishes that prompts are transformed according to 4 phases:

- Rapid arrangement of positions
- Stable, long-lived relations among prompts
- A transition phase where the model refines these relations
- Another phase of new rearrangements

I would rank the evidence for this claim as moderate. More specifically, the distinction between the phases is a bit fuzzy and hyper-parameter-dependent — for instance, the shape of the plots in Figure 3 changes significantly with alpha. Also, the paper only reports Birth's relative frequency plots for variations of Llama (do not consider other LLMs).

Also, Figure 2 aims to show "that a large amount of 1-dimensional holes are short-lived and that long-lived features appear after the first half of the model". First, it is unclear if this is the case  --- e.g., the number of tuples of persistence = 5 seems similar for l=5 and l=25 (or persistence = 10 for l=5 and l=20). Also, in Appendix D1, I expected to see the same plots for different datasets and models. However, the paper shows persistence differences for different model combinations, and it is unclear if it should lead to the same conclusion.

**Essential References Not Discussed:**

Overall, the paper does a good job covering related literature. I have no additional recommendations.

**Experimental Designs Or Analyses:**

Please see my comments about the experiments in "Methods and Evaluation Criterion" and "Claims and Evidence".

**Methods And Evaluation Criteria:**

In the first set of experiments, the paper mainly considers variants of Llama. Thus, it is hard to check if the findings apply to different models. The proposed evaluation criteria seem sensible, but since they are related to the proposed descriptor, they are not standard in the literature. Also, the paper's analysis only leverages the last token embeddings as a surrogate for the latent representation of the whole prompt.

For the experiments on layer pruning, the paper considers two other LLMs (Mistral and Pythia) and three benchmarks. This is less than found in other papers on layer pruning like "ShortGPT: Layers in Large Language Models are More Redundant Than You Expect".

Finally, it seems the analysis relies on 1-dim topological descriptors despite the paper mentioning $p$-dim holes.

**Other Comments Or Suggestions:**

Minor issues:

- Line 117: the choice for (Gromov et al, 2024) and (Men at al, 2024) is not motivated.
- Based on Fig. 1, the range of $b,d$ should be {0, \dots, 2(N - 1)}.
- Is $l_i=i$ in all cases? If so, why use $l_i$?

**Other Strengths And Weaknesses:**

**Strengths**

*Novelty*: As far as I know, this is the first paper to propose using zig-zag PH to track the evolution of internal representations of LLMs and NNs in general.

*Relevance*: The paper tackles the very relevant problem of understanding how LLMs work.


**Weaknesses**:

*Presentation*: The authors could provide more details on the zig-zag algorithm in the Appendix. For instance, how does it decide which simplex dies when two simplices get merged in the subsequent layer? Does it matter?

In addition, the paper introduces a variant of the PI vectorization scheme. However, there is little discussion about the variant. In particular, the specific form of Eq. (4) is not properly motivated. In addition, why is "smoothness as a function of layers" important for the application at hand?

*Results on layer pruning*. Overall, the results regarding layer pruning are not impressive. Indeed, since the competitive methods seem simpler (angular distance and bi-score), I am not sure if one would prefer this approach.

**Questions For Authors:**

1. Do the experiments only consider 1-dim holes? Why?
2. Could the proposed approach be used to analyze (or prune) other models (e.g., CNNs)? Would it apply to settings where the embedding dimension changes across layers?
3. Why do the proposed model and baselines only cut consecutive layers (see Table 2)?
4. What novel insights does this paper bring to the community? In other words, how does this paper contribute to advancing our understanding of LLMs that were not present in prior works?
5. How does the computational cost of the proposed method compare to those of (Gromov et al, 2024) and (Men at al, 2024)?

**Relation To Broader Scientific Literature:**

In Section 4.3, the paper briefly mentions how its findings (the identified four phases) can be related to previous literature/findings.

For instance, phase 1 has been related to local contextualization (Lad et al, 2024) and increased dimensionality (Valeriani et al, 2023). Phase 2 may be related to the decreasing dimensionality found in (Cheng et al., 2024). Overall, I found the discussion in Section 4.3 shallow.

It would be good if the authors could provide a summary of new insights and how it advances our understanding of LLMs.

**Theoretical Claims:**

The paper does not make theoretical claims.

---

> ### Author Rebuttal · Authors · 2025-03-31
>
> We thank the reviewer for their insightful feedback. They recognize the novelty of our approach to analyzing LLMs and its importance in enhancing understanding of these models. They also acknowledge a proper review of existing literature.
> As for feedback on points to improve, we reply element-wise below:
>
> > the shape of the plots in Figure 3 changes significantly with alpha
>
> The variability in alpha is a feature of the method since it allows to explore different regimes (i.e. short-lived vs long-lived features). This is explained in the paragraphs where phases are described. As a side note, we show that this behavior is consistent across models in App. D.
>
> > the paper only reports Birth's relative frequency plots for variations of Llama (do not consider other LLMs). [...] Also, in Appendix D1, I expected to see the same plots for different datasets and models.
>
> We have computed these plots, but did not add them to the manuscript. They can be accessed here: https://anonymous.4open.science/r/conferenceProject-019A/src/plots/Rebuttal-plots.md
>
> > [...] it is unclear if this is the case --- e.g., the number of tuples of persistence = 5 seems similar for l=5 and l=25 (or persistence = 10 for l=5 and l=20).
>
> We agree that this statement is hard to confirm by eye in Fig. 2. We are open to rephrase it as: “Figure 2 shows that features born after the first half of the model’s depth have a higher tendency to be long-lived with respect to features born earlier on.”
>
> > [..] This is less than found in other papers on layer pruning like "ShortGPT: Layers in Large Language Models are More Redundant Than You Expect".
>
> We have run additional benchmarks, results included here:  https://anonymous.4open.science/r/conferenceProject-019A/src/plots/Rebuttal-plots.md.
>
> >how does it decide which simplex dies when two simplices get merged in the subsequent layer? Does it matter?
>
> The topological feature that dies is uniquely determined by the decomposition of the zigzag module provided in Equations 11-12, removing any ambiguity in its identification.
>
> > [..] the specific form of Eq. (4) is not properly motivated. In addition, why is "smoothness as a function of layers" important for the application at hand?
>
> PIs are usually linked to a choice of kernel. In our case, the PD is already amenable to a density grid, the value of density being feature counts. Intersection layers are only necessary for defining the filtration (e.g. one could use the union equivalently). This is desirable since we are interested in studying statistically the LLM layers.
> The smoothness is related to smoothing formigram (Prop. C.2) in https://arxiv.org/abs/1712.04064 where they smooth barcodes defined on X (all the layers including intersection our case) to barcodes on S(X) (just the LLM layers)
>
> >Overall, the results regarding layer pruning are not impressive. Indeed, since the competitive methods seem simpler (angular distance and bi-score), I am not sure if one would prefer this approach.
>
> See a reply to a similar feedback in response to reviewer zXjm.
>
> Replies to minor issues:
>
> >the choice for (Gromov et al, 2024) and (Men at al, 2024) is not motivated.
>
> These methods were chosen as it is easy to compare with ours, since the algorithms require as an input the number of blocks to be removed, which is what our criterion based on inter-layer persistence outputs.
>
> >Based on Fig. 1, the range of (b,d) should be {0, \dots, 2(N - 1)}.
>
> Correct, this is a typo caused by Python notation. In fact, there are 2N-1 snapshots.
>
> >Is $l_i =i$ ? in all cases? If so, why use $l_i$?
>
> The notation $\ell_i$ was used to be suggestive of the fact that the indices refer to layers.
>
> Replies to “Questions for Authors”:
>
> 1. As noted at the beginning of Sec. 4.2, experiments are performed for 0,1,2,3-dim holes, however only 1-dim holes have large number counts so that a statistical study is stable.
>
> 2. Yes, the method can be used to analyze models with variable embedding dimension (e.g. CNNs), this is mostly because our method is distance-based and we can expect neighborhoods to shift smoothly in CNNs (see eg https://arxiv.org/pdf/2007.03506)
>
> 3. Even though our method allows us to cut non-consecutive layers, our results are in line with the compared methods cutting consecutive layers. If tested on more specialized prompts, our algorithm might cut non-consecutive layers for a low enough threshold (see e.g. Java code in Fig. 9).
>
> 4. The key point of our work is to build an interpretable framework that allows studying the internal reps of NNs as a whole system, rather than collecting summaries of snapshots and then combining them a posteriori. This is important in for interpreting different phases of prompt processing, as argued in our work.
>
> 5. The computational cost of our method is higher than other methods, as detailed in the reply to 4Kx7. While it is an important worry for scaling up our analysis, for this specific task, it is not an issue given pruning is performed only once.

---

### Official Review · Reviewer_zXJm · 2025-03-13

**Overall Recommendation:** 3

**Summary:**

The authors introduce the concept of Zigzag persistence from topological data analysis to understand how features evolve through layers. The authors aim to offer a statistical perspective on how prompts are rearranged and their relative positions changed in the representation space, providing insights into the system’s operation as an integrated whole.

**Claims And Evidence:**

The idea of using zigzag filtration to understand the internal representation of LLM is novel, and the authors present the concept in an accessible way to readers.

**Essential References Not Discussed:**

NA.

**Experimental Designs Or Analyses:**

The experiment results show that persistent topological features and their similarities remain consistent across various models, layers, and hyperparameter settings within the framework, indicating a level of universality in the topological structure of LLM representations.

**Methods And Evaluation Criteria:**

This paper focuses more on explaining the idea rather than evaluation. The authors evaluate their methods with four smaller, open-sourced models.

**Other Comments Or Suggestions:**

NA

**Other Strengths And Weaknesses:**

The strength is the novelty of this work; the authors apply this new idea to understand the consistency and features flow between different layers. The weakness is that the paper is proof-of-concept type, so the experiment result is still limited.

**Questions For Authors:**

My biggest question is why the idea of Zigzag is important/interesting to the ML community. I'm not an expert in TDA, but this looks like a niche concept to me.
The improvement in the accuracy of the authors' method compared to the pruning method looks marginal. Is it possible to report SE so the interpretation of the results can be more robust? Also the computational cost of the authors' method seems expensive.

**Relation To Broader Scientific Literature:**

See my questions for authors.

**Theoretical Claims:**

This paper is heavily dependent on the idea of zigzag in TDA. There's no proof.

---

> ### Author Rebuttal · Authors · 2025-03-31
>
> Reply to zXjm
>
> We thank the reviewer for their useful feedback. They highlight the novelty of using zigzag persistence from topological data analysis to understand feature evolution in LLMs, appreciating the accessible presentation. They raise a few concerns, which we address below:
>
> > My biggest question is why the idea of Zigzag is important/interesting to the ML community. I'm not an expert in TDA, but this looks like a niche concept to me
>
> Within the TDA community, there exist several works addressing time-varying point clouds. The reason for the relatively slower progress in this field has been mainly computational: in a similar way as in multi-persistence, the complexity grows rapidly when adding filtration parameters, and real-world applications are harder to implement.  Nevertheless, recent advances in fast algorithms have allowed these methods to be applied more widely. In the specific case of Zigzag persistence, the fast zigzag algorithm of [1] allowed us to run the pipeline on large and high-dimensional datasets (see reply to 4Kx7 for computational complexity).
> Consequently,  these algorithms have become feasible to apply in the framework of interpreting dynamical changes in internal representations of neural networks.
>
> In short, we would argue that recent computational advances in analysing time-varying point clouds with TDA approaches have greatly increased relevance for the broader ML community.
>
> > The improvement in the accuracy of the authors' method compared to the pruning method looks marginal
>
> As the review remarks earlier on, as the work’s main objective is to connect a relatively underexploited mathematical algorithm (zigzag) to LLMs interpretability, the scope of the layer pruning analysis is to show that our method can be used on a downstream task with performances on par with state of the art methods.
>
> > Is it possible to report SE so the interpretation of the results can be more robust?
>
> There is relatively little variability in the benchmarks used for layer pruning, as it comes from boostrapping. Consequently, our computation reveals a standard deviation of ~1e-4. We note that whenever the accuracy score is the same for our work and the cited references, the results are exactly equal, i.e. the methods suggest pruning the same layers.
>
> >Also the computational cost of the authors' method seems expensive.
>
> We refer to our reply to 4Kx7 for computational costs of the algorithm. As for the specific comparison to the  layer pruning methods cited in our work, our algorithm is relatively slower, but it contains strictly more information (beyond layer pruning), as it analyzes all layers at the same time, rather than considering pairwise comparisons among layers.
>
>
>
> [1] Dey, T. K. and Hou, T. Fast Computation of Zigzag Persistence. In Chechik, S., Navarro, G., Rotenberg, E., and Herman, G. (eds.), 30th Annual European Symposium on Algorithms (ESA 2022), volume 244 of Leibniz International Proceedings in Informatics

---

### Official Review · Reviewer_4Kx7 · 2025-03-21

**Overall Recommendation:** 3

**Summary:**

This work introduces a framework for applying the topological descriptor zigzag persistence to analyze the internal representations of large language models (LLM). The experiments are conducted to evaluate the LLM models (Llama2-7B, Llama3-8B, Mistral 7B, and Pythia 6.9B), which demonstrates the effectiveness of the proposed framework in understanding the internal dynamics of LLMs and its practical utility in tasks like layer pruning.

**Claims And Evidence:**

The authors provide experiments, statistical descriptors, and visualizations to validate the proposed framework.

**Essential References Not Discussed:**

Dynamic Persistence [1], related to zigzag persistence, should be added in references and compared with.

[1] Kim, Woojin, and Facundo Mémoli. "Spatiotemporal persistent homology for dynamic metric spaces." Discrete & Computational Geometry 66 (2021): 831-875.

**Experimental Designs Or Analyses:**

The effectiveness of zigzag PD and the effective persistence image should be evaluated further.

**Methods And Evaluation Criteria:**

The proposed method does make sense.

**Other Comments Or Suggestions:**

The runtime efficiency evaluation for the proposed method is missing, which is quite desirable to know the efficiency performance of the proposed method.

**Other Strengths And Weaknesses:**

Strengths:
This work provides a novel approach on combining zigzag persistence, a kind of topological descriptor, with LLM analysis, for tasks such as model interpretability, and the result illustrates the efficacy of the proposed method.


Weaknesses:
1. The integration of the proposed method into filtration, and the tracking of the evolution of internal representations both are desired, which deserve to be appended for further validating the proposed zigzag persistence based method.
2. The comparison to related TDA approach is insufficient.

**Questions For Authors:**

How does the proposed method compare to dynamic persistence?

**Relation To Broader Scientific Literature:**

Relation to TDA,  neural network interpretability on large language models, with the focus on on neural network interpretability by introducing topological descriptors that capture the stability and evolution of prompt relations across layers.

**Theoretical Claims:**

Just briefly check the proofs, no significant issues found.

---

> ### Author Rebuttal · Authors · 2025-03-31
>
> Reply to 4Kx7
>
> We thank reviewer 4Kx7 for their useful feedback. While recognizing the novelty of our approach in combining zigzag persistence for LLM interpretability, the reviewer correctly points out that our work should have included a broader discussion of similar TDA approaches. We agree with this comment and recognize it would improve the effectiveness of our work. Here we reply to direct points raised by the referee in this regard, and we would be open to including part of this discussion in an edited version of the manuscript
>
> > Dynamic Persistence [1], related to zigzag persistence, should be added in references and compared with.
>
>
> The reference is already cited in the main text as “Kim and Memoli, 2021.”
>
> > The integration of the proposed method into filtration, and the tracking of the evolution of internal representations both are desired, which deserve to be appended for further validating the proposed zigzag persistence based method.
>
> Given the large number of points considered in this analysis, we adopt a statistical approach to study the effect of layer transformations on internal representations, where the identity of individual points is not essential. The idea of tracking the zigzag representatives using [3] could certainly be investigated. Testing this pipeline on an example in a more controlled setting would be useful in validating the proposed zigzag framework and it would allow in principle to track more closely the evolution of features across layers. We deserve these studies for future work.
>
> >The comparison to related TDA approach is insufficient, How does the proposed method compare to dynamic persistence?
>
> Here are a few points of comparison:
>
> 1. We choose kNN filtration (whereas it is Rips in [1]) since it is more suitable to high dimensional data, especially for LLM reps. The stability of kNN filtrations is discussed in [4] in the context of persistent homology, we recognize that we missed this reference in the submitted manuscript, we would be open to add it in an edited version.
> 2. [1] varies both a time and a scale parameter whereas we fix the scale. Their main summary statistic (the rank invariant) involves calculating a 6-dimensional summary statistic (4 across layers and 2 across scale) which can make analysis hard for both computational reasons as well as finding the informative summary statistic. Nevertheless, varying also the scale parameter is definitely worth investigating in this context, and the techniques in [1] would be a starting point for implementing it.
> 3. The work [5] from the same authors is more related to our work since the maximal group diagram and the persistence clustergram described in the above paper are "annotated" (with the representative topological features) barcodes. In this work, they fix a scale similar to our case.
>
> >The runtime efficiency evaluation for the proposed method is missing, which is quite desirable to know the efficiency performance of the proposed method.
>
>
> A broad assessment of computational efficiency is present in the current version (Appendix B).  We agree this is an important point to address, thus here are more details on it: the theoretical complexity of FastZigZag is O($m^{\omega}$) with $\omega < 2.37286$ for a sequence of $m$ deletion and additions that is an advancement from previous algorithms with a cubic complexity, which allowed us to do our experiments with point clouds of 10k points over ~30 layers. More details on the algorithm are described in the paper by Dey and Hou [2].
> As for the knn graph, its creation with a greedy algorithm is O($n^2$) with n being the data points. In total, we have a O($n^2 * N_{layers}$) + O($m^{\omega}$). We would be open to adding this discussion in an edited version.
>
>
> [1] Kim, Woojin, and Facundo Mémoli. "Spatiotemporal persistent homology for dynamic metric spaces." Discrete & Computational Geometry 66 (2021): 831-875.
>
> [2] Dey, T. K. and Hou, T. Fast Computation of Zigzag Persistence. In Chechik, S., Navarro, G., Rotenberg, E., and Herman, G. (eds.), 30th Annual European Symposium on Algorithms (ESA 2022), volume 244 of Leibniz International Proceedings in Informatics
>
> [3] Dey, Tamal K., Tao Hou, and Dmitriy Morozov. "A Fast Algorithm for Computing Zigzag Representatives." Proceedings of the 2025 Annual ACM-SIAM Symposium on Discrete Algorithms (SODA). Society for Industrial and Applied Mathematics, 2025.
>
> [4] Le, Minh Quang, and Dane Taylor. "Persistent homology with k-nearest-neighbor filtrations reveals topological convergence of PageRank." Foundations of Data Science 7.2 (2025): 536-567.
>
> [5] Kim, Woojin, and Facundo Mémoli. "Extracting persistent clusters in dynamic data via möbius inversion." Discrete & Computational Geometry 71.4 (2024): 1276-1342.

---

### Decision · Program_Chairs · 2025-05-01

**Decision:**

Accept (poster)

**Comment:**

The paper uses zigzag persistence to track overlapping homological features in adjacent layers as data passes through an LLM. The authors analyze various statistics derived from the resulting persistence diagrams to characterize the internal dynamics of LLM as well as apply the resulting observations for pruning. Other than questions about exposition of the relatively obscure concept of zigzag persistence, the reviewers raised concerns about the marginal improvement in the authors' pruning methods, compared to existing techniques. The authors offered a rebuttal that pruning is only meant to illustrate that it's possible to use the ideas in the paper for downstream analysis; the main contribution is in offering a way to use a mathematical technique of zigzag persistence. The reviewers were satisfied with the rebuttal and raised some of their scores.

Please note: The discussion of FastZigzag algorithm in your rebuttal is misleading.
> We agree this is an important point to address, thus here are more details on it: the theoretical complexity of FastZigZag is $O(m^\omega)$ with $\omega < 2.37286$, for a sequence of $m$ deletion and additions **that is an advancement from previous algorithms with a cubic complexity**, which allowed us to do our experiments [...]. More details on the algorithm are described in the paper by Dey and Hou [2]. [...] We would be open to adding this discussion in an edited version.

This statement is imply not true. Not only has computation of zigzag persistence been known to be in matrix multiplication time [Milosavljevic et al., 2011], the argument in FastZigZag paper for this complexity simply appeals to that same paper. Moreover, the implementation of FastZigZag does not use matrix multiplication time algorithms. That is not the explanation for why it is faster in practice.

All that is to say, please don't add these comments to the paper, as you volunteer in the rebuttal.